# Thyroid Hormone Receptor Agonistic and Antagonistic Activity of Newly Synthesized Dihydroxylated Polybrominated Diphenyl Ethers: An In Vitro and In Silico Coactivator Recruitment Study

**DOI:** 10.3390/toxics12040281

**Published:** 2024-04-11

**Authors:** Mengtao Zhang, Jianghong Shi, Bing Li, Hui Ge, Huanyu Tao, Jiawei Zhang, Xiaoyan Li, Zongwei Cai

**Affiliations:** 1State Environmental Protection Key Laboratory of Integrated Surface Water-Groundwater Pollution Control, School of Environmental Science and Engineering, Southern University of Science and Technology, Shenzhen 518055, China; 11652003@mail.sustech.edu.cn (M.Z.); geh@sustech.edu.cn (H.G.); 11850006@mail.sustech.edu.cn (H.T.); 11750007@mail.sustech.edu.cn (J.Z.); 2China State Key Laboratory of Environmental and Biological Analysis, Department of Chemistry, Hong Kong Baptist University, Hong Kong, China; zwcai@hkbu.edu.hk; 3Institute of Environment and Ecology, Shenzhen International Graduate School, Tsinghua University, Shenzhen 518055, China; bingli@sz.tsinghua.edu.cn (B.L.); lixiaoyan@sz.tsinghua.edu.cn(X.L.)

**Keywords:** DiOH-PBDEs, thyroid hormone receptors, coactivator recruitment assay, agonistic or antagonistic activities

## Abstract

Dihydroxylated polybrominated diphenyl ethers (DiOH-PBDEs) could be the metabolites of PBDEs of some organisms or the natural products of certain marine bacteria and algae. OH-PBDEs may demonstrate binding affinity to thyroid hormone receptors (TRs) and can disrupt the functioning of the systems modulated by TRs. However, the thyroid hormone disruption mechanism of diOH-PBDEs remains elusive due to the absence of diOH-PBDEs standards. This investigation explores the potential disruptive effects of OH/diOH-PBDEs on thyroid hormones via competitive binding and coactivator recruitment with TRα and TRβ. At levels of 5000 nM and 25,000 nM, 6-OH-BDE-47 demonstrated significant recruitment of steroid receptor coactivator (SRC), whereas none of the diOH-PBDEs exhibited SRC recruitment within the range of 0.32–25,000 nM. AutoDock CrankPep (ADCP) simulations suggest that the conformation of SRC and TR–ligand complexes, particularly their interaction with Helix 12, rather than binding affinity, plays a pivotal role in ligand agonistic activity. 6,6′-diOH-BDE-47 displayed antagonistic activity towards both TRα and TRβ, while the antagonism of 3,5-diOH-BDE-100 for TRα and TRβ was concentration-dependent. 3,5-diOH-BDE-17 and 3,5-diOH-BDE-51 exhibited no discernible agonistic or antagonistic activities. Molecular docking analysis revealed that the binding energy of 3,3′,5-triiodo-L-thyronine (T3) surpassed that of OH/diOH-PBDEs. 3,5-diOH-BDE-100 exhibited the highest binding energy, whereas 6,6′-diOH-BDE-47 displayed the lowest. These findings suggest that the structural determinants influencing the agonistic and antagonistic activities of halogen phenols may be more intricate than previously proposed, involving factors beyond high-brominated PBDEs or hydroxyl group and bromine substitutions. It is likely that the agonistic or antagonistic propensities of OH/diOH-PBDEs are instigated by protein conformational changes rather than considerations of binding energy.

## 1. Introduction

Polybrominated diphenyl ethers (PBDEs) are a series of halogenated flame retardants that have been widely used in wire insulation, furniture, textiles, and electronic devices [1,2]. The commercial usage of penta-BDE, octa-BDE, and deca-BDE has been subject to regulation under the Stockholm Convention on Persistent Organic Pollutants since 2009 [3,4,5], aiming to curtail and limit the utilization of commercial PBDE products. PBDEs can be metabolized to hydroxylated polybrominated diphenyl ethers (OH-PBDEs) by cytochrome P-450-mediated biotransformation [6,7]. OH-PBDEs and dihydroxylated polybrominated diphenyl ethers (diOH-PBDEs) can also be the natural products of marine bacteria and sponges and/or their symbionts cyanobacteria [8]. The research conducted by Teuten et al. [9] analyzed the natural abundance of radiocarbon content of two MeO-PBDEs isolated from a True’s beaked whale (*Mesoplodon mirus*). It shows that MeO-PBDEs are naturally produced by ocean organisms such as algae and sponges and bioaccumulated in the beaked whale [9]. Additionally, Teuten et al. [10] identified MeO-PBDEs in pre-industrial whale oil, further supporting the natural origin of MeO-PBDEs. Agarwal et al. [11] reported marine bacteria as producers of OH-PBDEs and established a genetic and molecular foundation for their production. Now, it is widely believed the OH-PBDEs in the food chain mainly come from the biosynthesis of ocean microorganisms rather than commercial PBDE products [12] In our previous investigation, OH/diOH-PBDEs were also detected in sea fish available in markets, indicating the potential risk of human exposure to these compounds [13].

Several toxicity studies have indicated that OH-PBDEs possess the capability of competitive binding to thyroid hormone transporters and receptors owing to their structural resemblance to the thyroid hormone [14,15]. Experimental investigations have also unveiled the impact of OH-PBDEs on thyroid hormone metabolism in animals. Our study indicated that zebrafish exposure to 1 nM of 6-OH-BDE-47 can significantly elevate thyroid hormone levels [16]. Methimazole, a recognized disruptor of thyroid function, exhibited a notable association with thyroid disruption and developmental abnormalities, including delayed development, spinal curvature, and unexpanded swim bladders in zebrafish larvae [17]. In our previous study, exposure to 6-OH-BDE-47 and 6,6′-diOH-BDE-47 elicited similar malformations as those induced by methimazole [16]. The presence of the OH group has been linked to the bioavailability of PBDEs [18]. The markedly heightened toxicity of OH-PBDEs compared to PBDEs is primarily attributed to their increased hydrophilicity and binding affinity to transporter and receptor proteins. Therefore, we are intrigued to explore whether diOH-PBDEs, being more hydrophilic derivatives, exhibit stronger toxicity and receptor-binding effects. The study of diOH-PBDEs has been hindered by the absence of commercially available standards. Hence, dihydroxylated metabolites of PBDEs were synthesized as follows [13]: 3,5-dihydroxy-2,2′,4-tribromodiphenyl ether (3,5-diOH-BDE-17), 6,6′-dihydroxy-2,2′,4,4′-tetrabromodiphenyl ether (6,6′diOH-BDE47), 3,5-dihydroxy-2,3′,4,5′-tetrabromodiphenyl ether (3,5-diOH-BDE-51), and 3,5-dihydroxy-2,2′,4,4′,6-pentabromodiphenyl ether (3,5-diOH-BDE-100).

Although the OH-PBDE structures were similar, some showed agonist activity, and others showed antagonist activity [19,20]. Other structural analogs, such as diOH-PBDEs, may also have similar properties. The interaction of OH-PBDEs and the thyroid hormone receptors (TRs) is strongly associated with the toxicity of PBDEs. 4-OH-BDE-90 and 3-OH-BDE-47 exhibited antagonistic effects on the interaction between 3,3′,5-triiodo-L-thyronine (T3) (0.1 nM) and thyroid hormone receptors [21]. Notably, the presence of 4-HO-BDE-90 at a concentration of 10.0 μM exerted a substantial suppression on TRs-mediated transcriptional activity induced by T3. Four low-brominated OH-PBDEs (2′-OH-BDE-28, 3′-OH-BDE-28, 5-OH-BDE-47, 6-OH-BDE-47) were found to be TR agonists, while three high-brominated OH-PBDEs (3-OH-BDE-100, 3′-OH-BDE-154, 4-OH-BDE-188) showed antagonistic activity [20]. The interaction between T3 and TRs plays a crucial role in regulating the patterns of gene expression and maintaining equilibrium in energy storage and utilization. The activities of TRs are intricately linked to the functions of a steroid receptor coactivator (SRC) and a corepressor (NCoR) [14]. In the absence of a ligand, empty TRs recruit NCoR (Figure 1), which in turn recruit histone deacetylase (HDAC3) to repress expression via histone deacetylation. Binding of T3 leads to conformational changes in the TRs and dismissal of NCoR. The TRs-T3 complex can recruit the SRC (Figure 1), which includes histone acetyl and methyl transferase activity, thus facilitating the activation of general expression machinery and mRNA synthesis. Molecular dynamics simulations have unveiled that T3 binding triggers a cascade of conformational alterations, leading to the repositioning of Helix 12 (H12) and facilitating the recruitment of the coactivator [14]. Conversely, the interaction with antagonists augments the recruitment of the corepressor or impedes the binding of the coactivator. However, the diOH-PBDEs’ SRC recruitment disruption mechanism and the binding potency of diOH-PBDEs to the TR were limited.

In the present investigation, the agonistic and antagonistic activities of 3,5-diOH-BDE-17, 3,5-diOH-BDE-51, 3,5-diOH-BDE-100, 6-OH-BDE-47, and 6,6′-diOH-BDE-47 on the TRs were assessed utilizing an SRC recruitment assay and competitive binding assay. The binding of SRC with the TR–ligand complex was simulated by AutoDock CrankPep. Molecular docking and molecular dynamics simulations were performed to understand the structural basis of the experimentally observed TR-OH/diOH-PBDEs binding effect. Our results provided the first in vitro and in silico piece of evidence of the interaction between TR, coactivator, and diOH-PBDEs. The binding data obtained from both in vitro and in silico approaches hold significant implications for comprehending the toxicity and environmental risk posed by diOH-PBDEs.

## 2. Materials and Methods

### 2.1. Chemicals

Target compounds 3,5-diOH-BDE-17, 6,6′-diOH-BDE-47, 3,5-diOH-BDE-51, 3,5-diOH-BDE-100, and 6-OH-BDE-47 were synthesized in our laboratory (Figure 2). Dimethyl sulfoxide (DMSO) and 3,3,5-triiodo-l-thyronine (T3) with a purity of 99.0% and 98% were procured from Aladdin (Shanghai, China). The stock solutions were dissolved in DMSO and stored in amber glass vials at 4 °C. The concentrations for 3,5-diOH-BDE-17, 6,6′-diOH-BDE-47, 3,5-diOH-BDE-51, 3,5-diOH-BDE-100, 6-OH-BDE-47, and T3 were 2.5 mM. The LanthaScreen™ TR-FRET Thyroid Receptor alpha and beta Coactivator Assay kit was acquired from Invitrogen Corporation of Thermofisher Scientific (Carlsbad, CA, USA). The non-binding surface 384 well plates were obtained from Corning (New York, NY, USA).

### 2.2. Coactivator Recruitment Assay

Time-Resolved Fluorescence Resonance Energy Transfer (TR-FRET) was employed to assess the potential of OH/diOH-PBDEs as TR agonists or antagonists via the recruitment of a TR coactivator. Fluorescein–SRC (Sequence: LKEKHKILHRLLQDSSSPV) served as the coactivator in this investigation. The terbium label on the anti-GST antibody (TRα or TRβ receptor) is excited and emission at the wavelength 340 nm and 495 nm, respectively. The emission of 495 nm has a time delay. Meanwhile, 495 nm was the excitation wavelength for fluorescein–SRC, and 520 nm was the emission wavelength for fluorescein. The excitation of fluorescein depends on the distance between terbium and fluorescein, which signifies the distance between the TR receptor and the SRC.

The OH/diOH-PBDEs exhibit structural similarity to T3. If OH/diOH-PBDEs bind to the TR and display similar TR agonistic activity, fluorescein-SRC recruitment would occur, resulting in the emission of both 520 nm and 495 nm wavelengths. Conversely, if OH/diOH-PBDEs fail to bind with the TR or exhibit TR antagonistic effects, the recruitment of fluorescein-SRC would be hindered. Consequently, the energy emitted by terbium would not stimulate the distinct signal of fluorescein (520 nm) within SRC. The TR-FRET method can circumvent interference from compound auto-fluorescence or light scatter from precipitated compounds. The EnVision Multilabel Reader from PerkinElmer (Hopkinton, MA, USA) was recommended by the coactivator assay kit manual and utilized in this study. The delay time and integration time were set at 100 μs and 200 μs, respectively.

The coactivator recruitment experiment was conducted on a 384-well plate. The final concentrations of other reagents in the well were 0.5 nM TRβ LBD-GST, 200 nM Fluorescein-SRC, and 2 nM Tb anti-GST antibody. The incubation period was set to 2 h based on the kit manual. Experiments were conducted with three replicate wells for each sample in the 384-well assay plates, and three parallel samples were performed to ensure the accuracy and reproducibility of the results. The concentration of T3 ranged from 0.32 to 5000 nM and 0.032 to 1000 nM for TRα and TRβ for the dose–response curves of T3, respectively. In the TR coactivator recruitment assay to assess OH/diOH-PBDEs TR agonist activity, varying concentrations of ligands were added to the plate wells. The concentration of OH/diOH-PBDEs ranged from 0.32 nM to 25,000 nM, while T3 ranged from 0.08 nM to 1000 nM. For the study on competitive binding of OH/diOH-PBDEs with T3 on TR, OH/diOH-PBDEs were consistently added at a concentration of 25,000 nM to each well. T3 was within the concentration range of 1.6–2000 nM for the TRα group and 0.16–1000 nM for the TRβ group.

### 2.3. In Silico Simulations

The interaction between SRC and the TR–ligand complex was simulated using AutoDock CrankPep (ADCP, Version 1.0 rc1) [22]. ADCP is a specialized AutoDock docking engine designed specifically for peptide docking [23]. A Monte Carlo search algorithm is employed to fold the SRC while simultaneously optimizing its interaction with the TR–ligand complex. Each ADCP docking comprises a predetermined number of independent Monte Carlo searches known as replicas. In this investigation, 50 replicas were utilized, with each allocated 1 million Monte Carlo steps per amino acid of the SRC. Each search begins with an extended conformation of the SRC, generated from its sequence, randomly rotated, and randomly positioned within the docking box [23]. The dimensions of the docking box were determined based on the size of the SRC and the binding pocket of TRα and TRβ for the peptide.

Molecular docking was employed to simulate the binding affinity of TRα and TRβ with OH/diOH-PBDEs. During the molecular docking process, grid energy calculations and semi-flexible docking were performed using Autodock v4.2.6 (La Jolla, CA, USA) to generate the ligand–receptor complex. The optimal configuration was selected based on the conformation with the lowest binding energy. Molecular dynamics simulations were conducted to simulate the behavior of TRα and TRβ, along with the receptor–ligand complex, using GROMACS gromacs-2019.1. In this study, a simulation period of 10 nanoseconds was employed for the protein, with a time step of 0.2 femtoseconds. The protein–ligand interactions were depicted schematically using LIGPLOT v2.2.8. Visualization of the proteins was accomplished using UCSF ChimeraX v1.1 [24,25].

## 3. Results and Discussion

### 3.1. TR Coactivator Recruitment Assay for OH/diOH-PBDEs TR Agonists Activity

The in vitro assay for thyroid TR coactivator recruitment was employed to evaluate the TR activity of OH/diOH-PBDEs. In the absence of T3, TR formed complexes with nuclear receptor corepressors. Upon TR binding with T3, the nuclear receptor corepressors were released, thereby facilitating the recruitment of coactivator proteins such as SRC [26]. TRα exhibited a greater binding affinity to T3 than TRβ (Figure 3). The recruitment of TRα and TRβ to SRC was observed to be T3 dose-dependent, with respective half-maximal effective concentration (EC50) values of 63.85 nM and 157.19 nM. In the ^125^I-T3 TR competitive binding assay, the dissociation constants (Kd) of TRα-T3 and TRβ-T3 were measured as 0.058 nM and 0.081 nM [27], indicating a higher binding affinity of TRα to T3 than TRβ. Ocasio and Scanlan employed ^125^I-T3 as a probe that specifically binds to TR-LBD at the T3-binding site. Competitive binding of T3 or a ligand with TR-LBD displaces the probe from the receptor. Levy et al. reported an EC50 of 124 nM for TRα SRC2 recruitment, which is consistent with the findings of this study [28]. It is essential to acknowledge that the EC50 values may be influenced by variations in the binding assay protocol. As the competitive binding assay protocol may not capture the complete conformational changes in TR or the recruitment of SRC, the resulting binding EC50 could be lower than that obtained in this study. Based on Figure 3, the peak concentrations of T3 for TRα and TRβ approached but did not reach a maximum 520/495 signal. This outcome could potentially impact the precision of the EC50 values. However, despite setting the maximum concentration of OH/diOH-PBDEs in Figure 4 close to their maximum solubility in water, a complete dose–response curve for OH/diOH-PBDEs was still not achieved. Consequently, conducting a comparison of EC50 values between OH-PBDEs and T3 is infeasible, and the relatively lower accuracy of the EC50 values for T3 can be considered acceptable.

In the TRα SRC recruitment assay, only 6-OH-PBDEs at concentrations of 5000 nM and 25,000 nM exhibited significant SRC recruitment, as illustrated in Figure 4A. None of the other four diOH-PBDEs displayed SRC recruitment activity. Specifically, within the TRα group, 6-OH-BDE-47 at concentrations of 5000 nM and 25,000 nM demonstrated considerable SRC recruitment (Dunnett’s test, *p* = 0.008, *n* = 3; *p* = 0.001, *n* = 3), as depicted in Figure 4A. In the TRβ group, 6-OH-BDE-47 at a concentration of 25,000 nM exhibited notable SRC recruitment (Dunnett’s test, *p* = 0.01, *n* = 3), as shown in Figure 4B. None of the remaining four diOH-PBDEs exhibited SRC recruitment potency in either the TRα or TRβ coactivator assay groups (Figure 4). At the concentration of 25,000 nM, the 520/495 signal ratio for 6-OH-BDE-47 was 0.95 and 1.0 for the TRα and TRβ groups. However, these signals were markedly weaker compared to the T3-induced response at a concentration of 1000 nM. Therefore, 6-OH-BDE-47 was considered to be a weak TR agonist. In Ren et al.’s study (2013), four low-brominated OH-PBDEs (2′-OH-BDE-28, 3′-OH-BDE-28, 5-OH-BDE-47, 6-OH-BDE-47) were found to be TR agonists. Disruption of functions of thyroid hormone modulated pathways by OH-PBDEs was evaluated by assays of competitive binding, coactivator recruitment, and proliferation of GH3 cells in Chen et al. study [14]. Some OH-PBDEs were able to bind to TR with moderate affinities but were not agonists. The reported study and our data indicated that only a part of the OH/diOH-PBDEs showed TR agonist activities in the coactivator recruitment assay. In GH3 proliferation assays, 13 out of 16 HO-PBDEs were antagonists for the thyroid hormone [14]. Hofmann et al. constructed a TH-responsive luciferase-based reporter plasmid and established a reporter gene assay using the human hepatocarcinoma cell line HepG2 as the host system [29]. TBBPA is a bromine-generation retardant similar to PBDEs in this study. No adverse effects of TBBPA in concentrations up to 10 μM were detected in the HepG2 cell viability assay. However, TBBPA exhibited a substantial reduction in DIO1 expression at a concentration of 10 μM. This suggests that despite the absence of discernible thyroid hormone receptor activity in the testing method, contaminants may still elicit variances in gene expression.

The binding of SRC with the TR–ligand complex was simulated by ADCP. Different from the ligand binding deep into the receptors, the peptides bind to the shallow surface of the receptors. Furthermore, extended amino acid chains possess a greater number of degrees of freedom. As a consequence, this study employed 50 replicas, each allocated 1 million Monte Carlo steps per amino acid. Data reduction is necessary when analyzing the docking results. The term “cluster” in Table 1 refers to a group of similar molecular conformations that are identified during the analysis of docking results. The interaction between SRC and the TRα–ligand complex exhibited a range of −22.1 to −20.9 kcal/mol for best binding affinity and −21.1 to −18.0 kcal/mol for the cluster average binding affinity (Table 1). The association between SRC and the TRβ–ligand showcased a higher degree of binding affinity, ranging from −25.3 to −22.0 kcal/mol for best binding affinity and −23.4 to −19.4 kcal/mol for cluster average binding affinity (Table 1). Remarkably, the highest binding affinity was not observed in the T3 group. Nonetheless, only T3 and 6-OH-BDE-47 exhibited SRC recruitment and TR agonist activity (Figure 4). The binding affinity of SRC to the TR–ligand did not demonstrate a direct correlation with agonist activity.

The configurations of SRC engaging with TR–ligand complexes, as simulated by ADCP, are depicted in Figure 5. SRC’s binding with TR-T3 and TR-6-OH-BDE-47 (light blue and light purple) revealed a similar conformation for both TRα and TRβ (Figure 5A,C). TR consists of twelve α-helices (Helix 1–Helix 12), and SRC exhibited stronger interaction with Helix 12 in TR-T3 and TR-6-OH-BDE-47 than with other TR-diOH-PBDEs (Figure 5). The binding of an agonist to TR induces a conformational shift around Helix 12, resulting in heightened affinity for the coactivator peptide [14,20]. The binding of an agonist with TR triggers a conformational alteration around Helix 12, resulting in an increased affinity for the coactivator peptide [14,20]. This suggests that the conformation of SRC and TR–ligand complexes, particularly their interaction with Helix 12, plays a pivotal role in ligand agonistic activity rather than merely the binding affinity. The experimental 3D structure of TR binding with SCR2-2 was not reported. When considering only the top-ranking solution, the docking success rate of ADCP reached approximately 62% for the complexes [30]. The predicted 3D structure of TR–ligand–SRC by ADCP served as a supplementary tool for understanding the interaction between the nuclear receptor and coactivator. However, the experimental 3D structure of the complex remains imperative.

### 3.2. OH/diOH-PBDEs Competitive Binding with T3 for TR Antagonistic Activity Assay

Figure 4 depicts that the four diOH-PBDEs failed to induce SRC recruitment within the concentration spectrum of 0.32–25,000 nM. However, whether the incapacity of diOH-PBDEs to bind to TRα and TRβ is attributed to their antagonistic activity or merely their lack of binding capability remains ambiguous. To delve further into this, a TR-FRET coactivator assay was conducted to assess the competitive binding of T3 and OH/diOH-PBDEs (Figure 6). OH/diOH-PBDEs were consistently added at a concentration of 25,000 nM to each well. T3 ranged from 1.6 to 2000 nM for the TRα group and from 0.16 to 1000 nM for the TRβ group (Figure 6). The groups treated with 3,5-diOH-BDE-17 and 3,5-diOH-BDE-51 exhibited dose-signal curves remarkably akin to those of the T3-only group (Figure 6). This observation suggests that the competitive binding affinity was minimal at a concentration of 25,000 nM for 3,5-diOH-BDE-17 and 3,5-diOH-BDE-51. At lower T3 concentrations (below 40 nM), the T3 + 6-OH-BDE-47 groups exhibited a statistically significant difference compared to the T3-only group, indicative of 6-OH-BDE-47′s status as a weak TR agonist. However, at higher T3 concentrations, the disparity diminished. For T3 concentrations exceeding 40 nM, the 6,6′-diOH-BDE-47 group exhibited a marked difference compared to the T3-only group. 3,5-diOH-BDE-100 also showed antagonistic activity up to 200 nM for TRα and 20–100 nM for TRβ. This reveals that 6,6′-diOH-BDE-47 exhibited antagonistic activity for both TRα and TRβ, while 3,5-diOH-BDE-100 antagonistic for TRα and TRβ was concentration related. Ren et al. assessed the activity of OH-PBDEs as TR antagonists by examining the cell proliferation of a TR-dependent rat pituitary tumor cell line [20]. The study also indicated that high-brominated PBDEs, including 3-OH-BDE-100, exhibit antagonistic activity [20]. A CHO-K1 cell–based reporter gene assay demonstrated that 4-HO-BDE-90 has antagonistic activity against both TRα and TRβ [19]. Kojima et al. proposed that essential structural factors such as the 4-hydroxyl group and two bromine substitutions adjacent to the hydroxyl group on the phenyl group contribute to the antagonistic activity of OH-PBDEs for TR [19]. The antagonistic activity of 6,6′-diOH-BDE-47 and 3,5-diOH-BDE-100 in this study suggests that structural factors determining the agonistic and antagonistic properties of halogen phenols may be more intricate than those involving high-brominated PBDEs or hydroxyl group and bromine substitutions.

### 3.3. Molecular Docking and Molecular Dynamic Simulations of OH/diOH-PBDEs with TR

From the preceding outcomes, it can be found that the tested OH/diOH-PBDEs, with very similar chemical structures and only different in the degree of bromination or hydrogen groups, have different activities (agonistic or antagonistic) for TR. The interactions between OH-PBDEs and TR were scrutinized via molecular docking. Visual depictions of the TRα and TRβ binding complexes in the presence of T3 and OH/diOH-PBDEs are presented in Figure 6 and Figure 7. Binding parameters obtained from the Autodock analysis are listed in Table 2. The affinity of OH/diOH-PBDEs for TR appeared to be less potent compared to T3. The hydrophilic amino acid substitutions reside in the inner confines of the ligand binding domain, while the OH groups are positioned towards the entry point of this domain (Figure 7). T3 assumed an appropriate orientation upon binding in accordance with observations from the human thyroid hormone receptor [31]. Hydrogen bonding interactions were observed between T3 and the side chains of Arg87 (TRα), Met118 (TRα), Met313 (TRβ), Arg320 (TRβ), Asn331 (TRβ), and His435 (TRβ) (Table 2). His435 was also noted in other T3 hydrogen bonds in human thyroid hormone receptors [20]. The binding mechanism of OH/diOH-PBDEs exhibited significant parallels with that of T3. All OH/diOH-PBDEs under examination could fit within the TR binding pocket (Figure 7). However, distinct binding positions and geometries were evident between T3 and OH/diOH-PBDEs. The length of T3 extended to 12.37 Å, surpassing the range of OH/diOH-PBDEs (10.19–11.29 Å). T3 exhibited a deeper binding orientation within the pocket. The deeper binding of T3 with TR might be attributed to its relatively lower hydrophobic interactions, thereby resulting in a reduced steric effect.

The binding energy of T3 surpassed that of OH/diOH-PBDEs. The computed binding energies of T3 with TRα and TRβ were −10.51 and −11.75 kcal/mol, respectively (Table 2). The binding energies of OH/diOH-PBDEs with TRα ranged from −7.50 to −9.55 kcal/mol, and with TRβ ranged from −7.74 to −8.95 kcal/mol (Table 2). Notably, 3,5-diOH-BDE-17, 3,5-diOH-BDE-51, and 3,5-diOH-BDE-100 exhibited highly analogous chemical structures, differing mainly in their bromination degrees. Evidently, the binding energy trend demonstrated an increasing pattern with higher bromine content. Among the seven OH/diOH-PBDEs investigated in this study, 3,5-diOH-BDE-100, with the greatest number of bromine atoms, demonstrated the most favorable binding energy. This suggests that the hydrophobic interaction of OH/diOH-PBDEs with TR plays a pivotal role in determining the binding energy.

Both the conformational dynamics of TR-OH/diOH-PBDEs complexes and the positional dynamics of OH/diOH-PBDEs were acceptable. The root-mean-squared deviation (RMSD) was used as an indicator to monitor the conformational dynamic stability of the protein–ligand complex during the molecular dynamics simulations. In this study, the MD simulations were extended over a duration of 10 ns with a time step of 10 ps. R The RMSDs within the TRα–ligand complex surpassed those within the TRβ–ligand complex. It indicated that the TRα-OH/diOH-PBDEs complex exhibited heightened conformational dynamics in comparison to its TRβ-OH/diOH-PBDEs counterpart. The prevalence of hydrophobic amino acids within TRβ’s composition may contribute to the stability of the protein, as these hydrophobic amino acids are capable of reinforcing the protein’s structural integrity via hydrophobic interactions [32]. Furthermore, the relatively diminished occurrence of loops within TRβ’s structure could also be implicated in yielding lower RMSD values [33]. The entire TR-OH/diOH-PBDEs system reached an equilibrium state around the 2 ns mark (Figure 8). More specifically, in the context of the TRα-OH/diOH-PBDEs complex, equilibrium was achieved after approximately 5 ns, while within the TRβ-OH/diOH-PBDEs complex, equilibrium was reached at the 2 ns interval (Figure 8A,B). It is worth noting that OH/diOH-PBDEs exerted discernible influence over the conformational dynamics observed.

RMSDs of OH/diOH-PBDEs were calculated to assess the position dynamic of the ligand. The RMSD of OH/diOH-PBDEs binding to TRα exhibited a stabilizing trend within the initial 2 ns, indicative of a steady repositioning of OH/diOH-PBDEs during this timeframe (Figure 8C,D). The averaged RMSD values associated with OH/diOH-PBDEs interaction with TRα ranged from 0.04 to 0.12 nm. In contrast, the association of T3 with TRα demonstrated a relatively heightened positional dynamism, with an RMSD of 0.15 nm. Similarly, the amalgamation of OH/diOH-PBDEs with TRβ attained stability after a 2 ns period. The averaged RMSD values for OH/diOH-PBDEs binding to TRβ spanned from 0.04 to 0.18 nm. 3,5-diOH-BDE-17 had a relatively high position dynamic when combined with TRβ.

## 4. Conclusions

In the thyroid hormones SRC recruitment assay, only 6-OH-PBDEs at concentrations of 5000 nM and 25,000 nM exhibited significant SRC recruitment. None of the diOH-PBDEs elicited SRC recruitment within the range of 0.32–25,000 nM. ADCP simulations revealed that the binding affinity of SRC to the TR–ligand did not show a direct correlation with agonist activity. The conformation of SRC and TR–ligand complexes, particularly their interaction with Helix 12, may play a pivotal role in ligand agonistic activity. 6,6′-diOH-BDE-47 displayed antagonistic activity for both TRα and TRβ, while the antagonistic effect of 3,5-diOH-BDE-100 on TRα and TRβ appeared concentration-dependent. Molecular docking analysis indicated that the binding energy of T3 surpassed that of the OH/diOH-PBDEs. 3,5-diOH-BDE-100 exhibited the highest binding energy, whereas 6,6′-diOH-BDE-47 displayed the lowest. Intriguingly, both compounds demonstrated antagonistic attributes towards thyroid hormones. Conversely, 3,5-diOH-BDE-17 and 3,5-diOH-BDE-51, closely resembling 3,5-diOH-BDE-100 in chemical structure, albeit differing solely in the extent of bromination, exhibited no discernible antagonistic activities. Structural factors determining the agonistic and antagonistic properties of halogen phenols may be more complex than those proposed by previous studies on high-brominated PBDEs or hydroxyl group and bromine substitutions. Both the agonistic and antagonistic propensities observed in the study of OH/diOH-PBDEs imply that their activity is likely influenced by subtle protein conformational changes rather than considerations of binding energy. Future research should aim to deepen the understanding of the molecular mechanisms underlying the activity of OH/diOH-PBDEs and their interactions with thyroid hormone receptors, particularly focusing on their interactions with Helix 12. Understanding how these interactions influence ligand agonistic/antagonistic activity could provide valuable insights into the mechanisms underlying thyroid hormone modulation. Furthermore, cellular experiments and in vitro experiments associated with the thyroid system are necessary to elucidate the toxicity and exposure risk of OH/diOH-PBDEs.

## Figures and Tables

**Figure 1 toxics-12-00281-f001:**
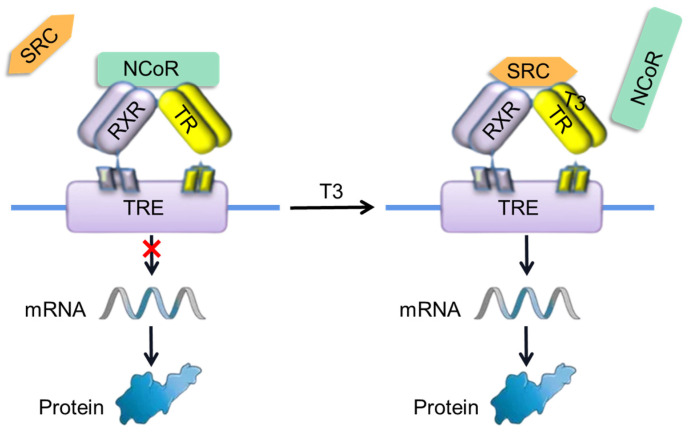
T3 binds to TRs to modulate gene expression in the nucleus. TRs heterodimerize with retinoid X receptor (RXR) on thyroid hormone response element (TRE) in the genome. In the absence of T3, the TRs recruit nuclear corepressors (NCoR), repress TRE, and affect the expression of TR target genes. TRs binding to T3 induces the dismissal of NCoR, recruitment of steroid receptor coactivator (SRC), and activation of the gene expression.

**Figure 2 toxics-12-00281-f002:**
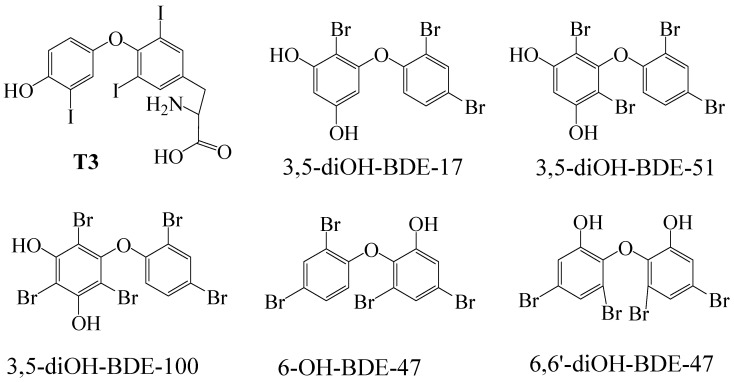
Chemical structures for T3 and OH/diOH-PBDEs used in the present study.

**Figure 3 toxics-12-00281-f003:**
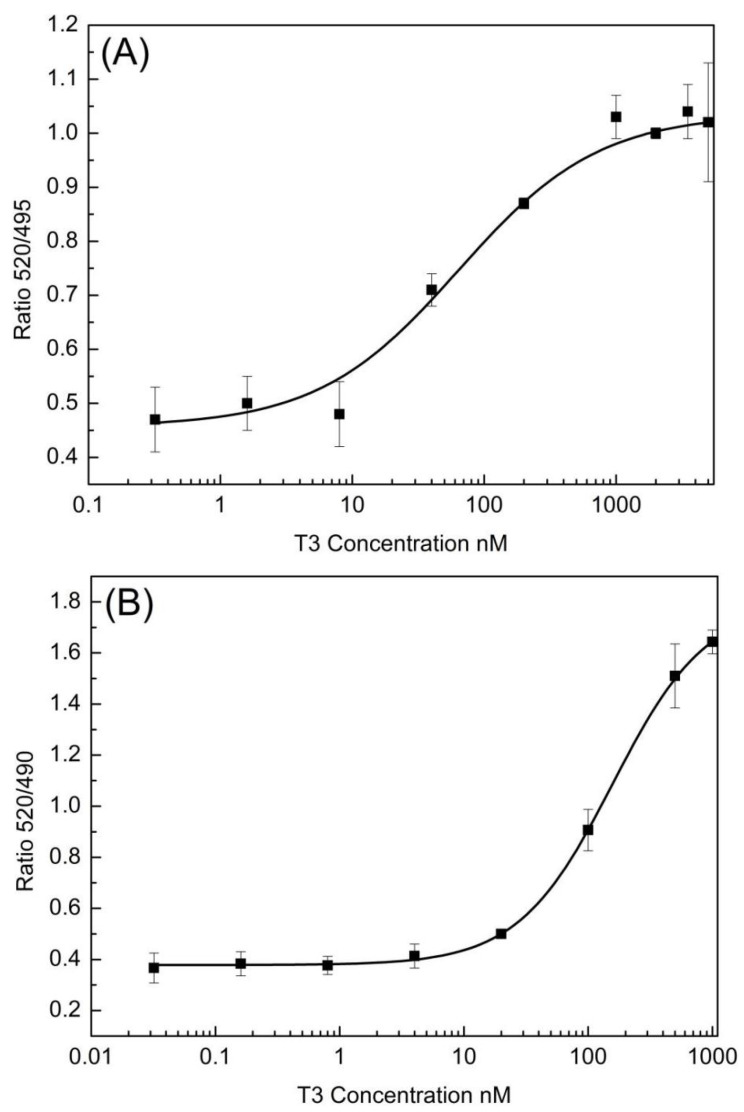
Dose–response curves of T3 for (**A**) TRα and (**B**) TRβ coactivator recruitment assay with serial dilution of agonist T3 for 2 h incubation. Values represent mean ± SD from three independent experiments performed in triplicates. Curves were fit using a sigmoidal dose–response equation in OriginLab Origin 8.1.

**Figure 4 toxics-12-00281-f004:**
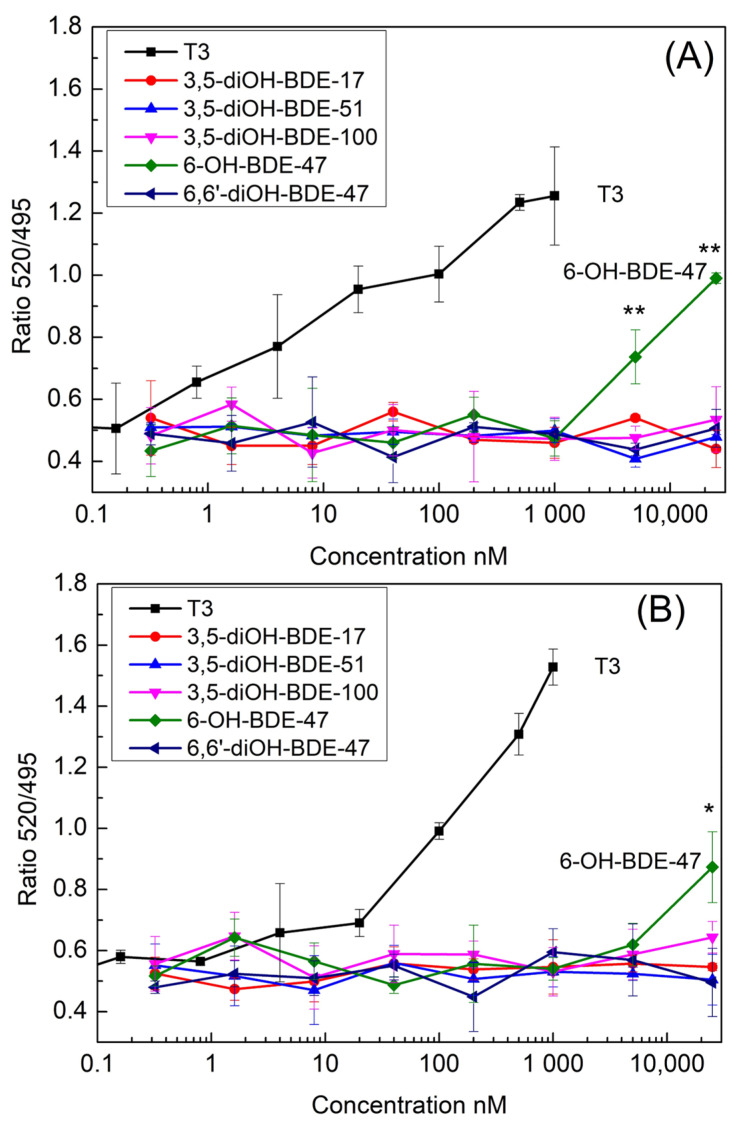
TR coactivator recruitment assay for OH/diOH-PBDEs TR agonist activity. (**A**) TRα and (**B**) TRβ binding to T3 or OH/diOH-PBDEs induces the recruitment of SRC and activation 520/495 signal. Values represent mean ± SD from three independent experiments performed in triplicates. One-way analysis of variance (ANOVA) with Dunnett’s test was applied for the statistical analysis. * *p* < 0.05 and ** *p* < 0.01 indicate significant differences between the exposure groups and control group.

**Figure 5 toxics-12-00281-f005:**
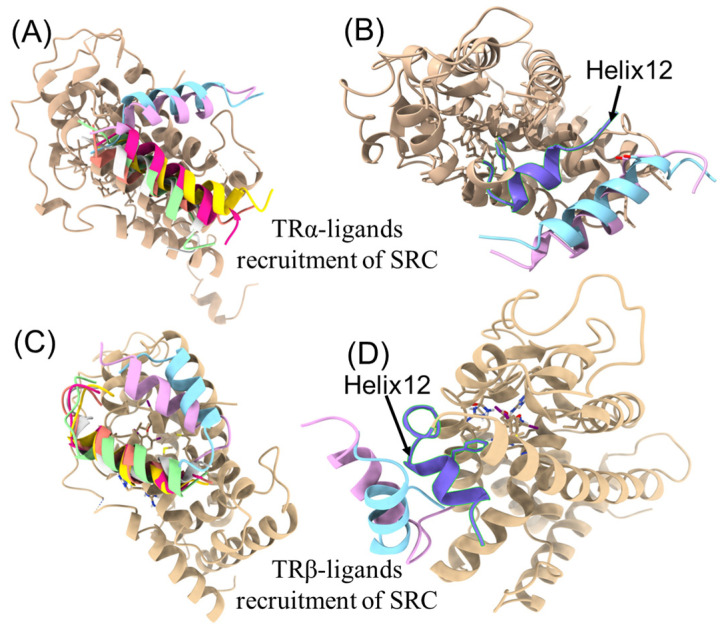
Binding of SRC with TR–ligand complex simulated by AutoDock CrankPep. In (**A**,**C**), light blue TR-T3-SRC, light purple TR-6-OH-BDE-47-SRC, light green, reddish brown, and magenta are TR-diOH-PBDE-SRC, and light yellow is without any ligand binding; (**B**,**D**) are TR-T3-SRC and TR-6-OH-BDE-47-SRC complex interaction with Helix 12.

**Figure 6 toxics-12-00281-f006:**
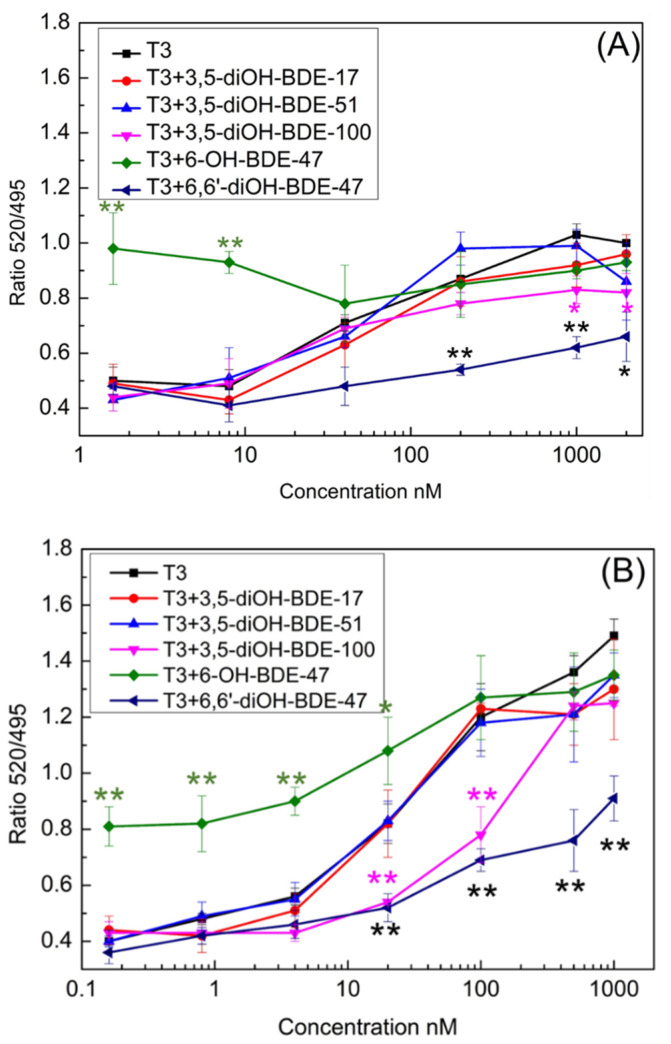
OH/diOH-PBDEs competitive binding with T3 for TR antagonistic activity assay. (**A**) TRα and (**B**) TRβ binding to T3 or OH/diOH-PBDEs induces the recruitment of SRC and activation 520/495 signal. OH/diOH-PBDEs were consistently at a concentration of 25,000 nM in each group. T3 was within the concentration range of 1.6–2000 nM for the TRα group and 0.16–1000 nM for the TRβ group. Values represent mean ± SD from three independent experiments performed in triplicates. One-way analysis of variance (ANOVA) with Dunnett’s test was applied for the statistical analysis. * *p* < 0.05 and ** *p* < 0.01 indicate significant differences between the exposure groups and control group.

**Figure 7 toxics-12-00281-f007:**
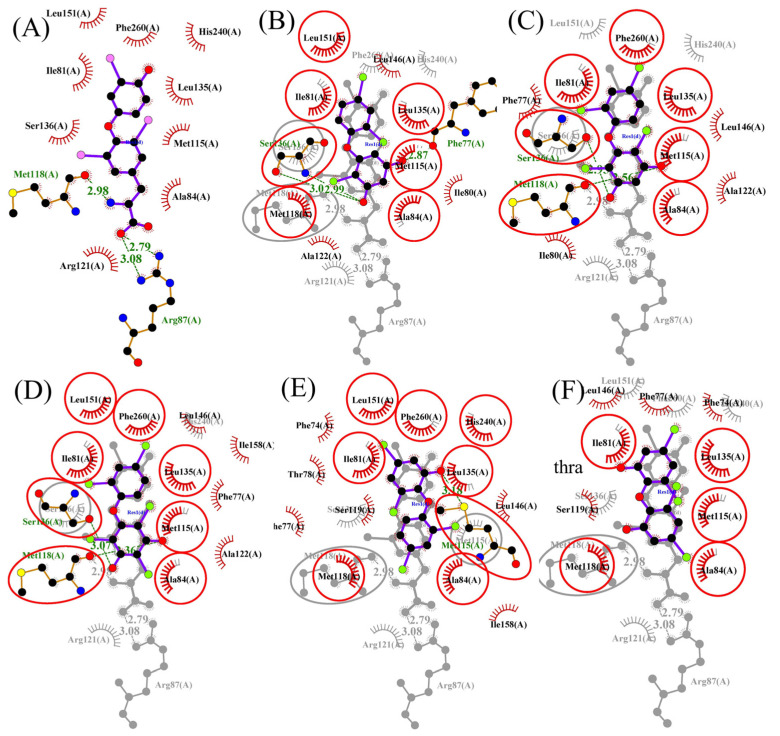
Ligand–protein interaction diagrams of TRα and TRβ with T3 (**A**,**G**) and T3 superposed diagrams with 3,5-diOH-BDE-17 (**B**,**H**), 3,5-diOH-BDE-51 (**C**,**I**), 3,5-diOH-BDE-100 (**D**,**J**), 6-OH-BDE-47 (**E**,**K**), and 6,6′-diOH-BDE-47 (**F**,**L**). Any conserved interactions are highlighted. The ligands and protein side chains are shown in ball-and-stick representation, with the ligand bonds colored purple. Hydrogen bonds are shown as green dotted lines, while the spoked arcs represent protein residues making nonbonded contacts with the ligand. The red circles and ellipses indicate protein residues that are in equivalent 3D positions when the two structural models are superposed.

**Figure 8 toxics-12-00281-f008:**
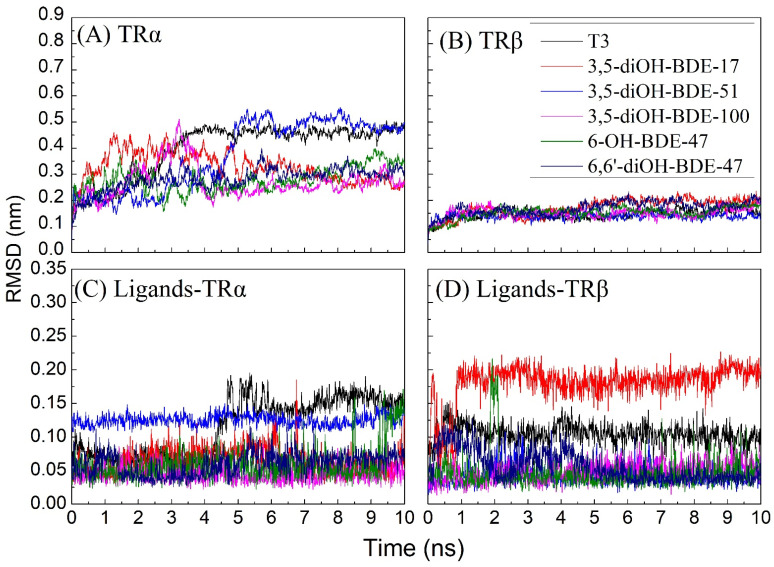
The root-mean-square deviation (RMSD) of carbon atoms (backbone) of (**A**) TRα and (**B**) TRβ OH/diOH-PBDEs and RMSD of OH/diOH-PBDEs in the (**C**) TRα and (**D**) TRβ monitored during a 10 ns MD simulation trajectory.

**Table 1 toxics-12-00281-t001:** Interaction of SRC and the TR–ligand complex simulated by AutoDock CrankPep.

Compound	TRα			TRβ		
	Best Binding Affinity (kcal/M)	Cluster Size	Cluster Average Binding Affinity (kcal/M)	Best Binding Affinity (kcal/M)	Cluster Size	Cluster Average Binding Affinity (kcal/M)
T3	−21.2	32	−19.3	−24.3	28	−21.2
3,5-diOH-BDE-17	−21.1	18	−19.3	−22.0	34	−19.9
3,5-diOH-BDE-51	−20.7	29	−19.0	−22.3	36	−19.4
3,5-diOH-BDE-100	−21.8	37	−19.3	−22.7	13	−22.2
6-OH-BDE-47	−21.5	10	−21.1	−25.3	32	−23.4
6,6′-diOH-BDE-47	−22.1	52	−19.6	−22.4	24	−21.5

**Table 2 toxics-12-00281-t002:** Molecular length and docking results of T3 and OH-PBDEs obtained for TRα and TRβ.

Compound	Length (Å)	TRα		TRβ	
		Docking Energy (kcal/M)	Hydrogen Bonding	Docking Energy (kcal/M)	Hydrogen Bonding
T3	12.37	−10.51	Arg87, Met118	−11.75	Met313, Asn331, Arg320, His435
3,5-diOH-BDE-17	10.21	−8.26	Ser136, Phe77	−8.01	Met313, Ser314, Met310
3,5-diOH-BDE-51	10.41	−8.73	Met118, Ser136	−8.35	Met313
3,5-diOH-BDE-100	11.20	−9.55	Met118, Ser136	−8.95	Met313, Asn331
6-OH-BDE-47	10.99	−8.25	Met115	−8.05	His435
6,6′-diOH-BDE-47	11.29	−7.50	no	−7.74	no

## Data Availability

The data presented in this study are available on request from the corresponding author.

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
