# Peer review of "Thyroid Hormone Receptor Agonistic and Antagonistic Activity of Newly Synthesized Dihydroxylated Polybrominated Diphenyl Ethers: An In Vitro and In Silico Coactivator Recruitment Study"

_toxics, 2024, doi:10.3390/toxics12040281_

Round 1

Reviewer 1 Report

Comments and Suggestions for Authors

Thyroid Hormone Receptor Agonistic and Antagonistic 2 Activity of Newly Synthesized Dihydroxylated 3 Polybrominated Diphenyl Ethers: An In Vitro and In Silico 4 Coactivator Recruitment Study 5 Mengtao Zhang 1,2,3, Jianghong Shi 1,*, Bin Li 1 , Hui Ge 1 , Huanyu Tao 1 , Jiawei Zhang 1 , Xiaoyan Li 3 6 and Zongwei Cai 2

Summary: This study investigates the effect of dihydroxylated polybrominated diphenyl ethers on the coactivator recruitment of thyroid hormone receptors alpha and beta (TRa and TRb). For this, they used a commercially available kit for coactivator recruitment and molecular docking in order to identify agonistic and antagonist effects. In principal, the study was carried out well. Its strength are the in situ docking studies, which bring novelty to the field. The conclusion made from the data are sound and might be interesting for the journals’ readers. However, some points are missed that need to be discussed more detailed/improved.

Major remarks:

·         The in vitro studies was carried out using a commercial kit. First question that comes to mind is why was the concentration range of T3 different between TRa and TRb? Especially since the potency for TRb is higher, the lower amount results in a cut off before saturation is reached. Here, an estimation of potency is questionable. The difference in concentration range is briefly mentioned in the materials and methods section, but no further explanation was given. In order to fully interpret the data concerning the compounds, it would be helpful to include some kind of negative control/blank. Compounds without agonistic effect would be similar to this control.

·         The recruitment kit is a pure biochemical assay that does not require cell-based expression. It measures the distance of SRC2-2 and TRa or TRb ligand binding domain (LBD) via TR-FRET. It should be discussed whether using only the LBD and the recruitment of a coactivator is different from studies using the whole receptor protein e.g. cell-based reporter gene assays that give information how a compound activates transcriptional activity (Hofmann et al, Toxicological Sciences, 2009).

·         Statistical methods are absent. What was used for non-linear regression model in Fig 2? What tests were used to determine significance? This also needs to be explained in the figure legends.

·         All figure legends concerning the in vitro studies are missing information (especially Fig 3 and Fig 4) and they fail to fully explain results. It should be apparent from the figure legend alone, what has been done and what can be seen. Also, it should include information about sample size and statistics.

Minor remarks:

·         Line 160: “Each sample was triplicate” is incomprehensible. Better to say “Each sample was measured in triplicates” or something alike.

·         Line 194f: SRC was mentioned here in full length (steroid receptor coactivator) for the first time. I would suggest adding the abbreviation in brackets, so the reader understands that this is identical to the aforementioned co activator. Best to use the same abbreviation throughout the manuscript.

·         Line 222f:  “Therefore, 6-OH-BDE-47 was considered to be weak TR agonists”, Should be “a” weak TR agonist”, Singular needs to be used if only one compound is discussed (agonist without the s).

·         Line 224ff: “in” missing before comparison. What is referred to by the “(Fig. 4A)”? The discussed study (Ren et al) or the presented study? “…, which consist with the 6-OH-BDE-47 in this study“ should be “…, which is consistent…”. Next sentence “…that no significantly (p<0.05) 226 Toxics 2024, 12, x FOR PEER REVIEW 7 of 16 agonists effect was observed><0.05) agonists effect was observed…” should be “… that no significant agonistic effect was observed”

·         Figure 4: Significance for 6-OH-BDE-47 could be marked by “*”, but this has to be mentioned in the legend and fully explained.

·         Line 233ff: “Different from the ligand binding deep into the receptors, the peptide binding to the shallow surface of the receptors” This sentence misses a verb all together. I suggest “… receptors, peptides are binding” but I’m not sure if this is what the authors wanted to say.

·         Line 287: too many spaces before “6,6'-diOH-BDE-47”

·         Line 293: Values at 2000 nM T3 at TRb was discussed, but could not be seen in the Fig 6 as again the concentration range was set to 1000 nM T3 for this receptor.

·         Line 300: “Compared the reported study on…” needs to be “Comparing the reported study to…”

·         Line 306f: “complicate” needs to be corrected to “complicated”

·         Figure 6: Legend is a bit better, though last sentence is missing words to be intelligible

·         Line 321: Fig 6 is mentioned though only Fig 7 depicted what was described.

·         Line 338: “was” is missing before observed

Comments on the Quality of English Language

English must be improved as stated in the comments above. I suggest all authors proof read the whole manuscript as language quality differs immensly between paragraphs.

Reviewer 2 Report

Comments and Suggestions for Authors

This type of study has been conducted in the past, so I hope there are clear differences in addition to new information regarding the substances tested.

In addition to T3, I believe it is necessary to include a clear positive control that can be compared to the PBDEs.

If structural similarity is important in the current study, I would be interested in seeing a description of the physicochemical properties of similar chemicals within the same group.

In Figure 7, the resolution is poor and the index position needs to be adjusted.

Figure 8. (B) Please review the legend line.

It would be helpful if you could provide five to ten references, more in the discussion section.

Comments on the Quality of English Language

Overall, I think it's well written, but please check again for typos.

Reviewer 3 Report

Comments and Suggestions for Authors

To the authors:

Page 1, Abstract first two lines you wrote: (Dihydroxylated polybrominated diphenyl ethers (DiOH-PBDEs) could be the metabolites of PBDEs of organism or the natural products of marine bacteria and algal.) It is better if it reads: 

(Dihydroxylated polybrominated diphenyl ethers (DiOH-PBDEs) could be the metabolites of PBDEs of some organisms or the natural products of certain marine bacteria and algae). 

In the abstract also, you have mentioned an abbreviation without defining it. Such as the (T3) in line 28 of the abstract. You mentioned it 11 times before you defined it in the material and method. Down on page 3! 

3, 3, 5-triodo-i-thyronine (T3) should be defined on page 1 in the abstract, and all other abbreviations. Unless you list all abbreviations at the bottom of the manuscript. 

All abbreviations need to be defined first, then re-used after it was defined and taken between (Parenthesis). 

Another question is the name correct? 3, 3, 5-triodo-i-thyronine (T3) or is it supposed to be 3, 3, 5-triiodo-l-thyronine (T3)? 

Another abbreviation not defined in the abstract is SRC2-2. What is that?  

Keyword: None given! 

Are the authors serious about their submission? 

Page 2, First few lines the authors wrote:

PBDEs can be metabolized to hydroxylated polybrominated diphenyl ethers (OH-PBDEs) by cytochrome P-450-mediated biotransformation (Mizukawa et al. 2015, Zheng et al. 2015). OH-PBDEs and dihydroxylated polybrominated diphenyl ethers (diOH-PBDEs) can also be the natural products of marine bacteria and sponges and/or their symbionts cyanobacteria (Agarwal et al. 2017), bacteria (Agarwal et al. 2014), red algal (Malmvärn et al. 2005), brown algal (Dahlgren et al. 2015, Goto et al. 2017) and green algal (Kuniyoshi et al. 1985) through oxidative dimerization of bromophenol compounds.

What kind of sentence this is? This runoff sentence do not have subject verb agreement, and no objective.

Instead of multiline sentences, (write short sentences to the point with subject, verb, and objective).

Line 53 the authors wrote:

It shows that these compounds were naturally produced in the beaked whale.

Where is the reference to this claim? Who and when the whales were beaked? Or Baked.

Page 3 line 129 under material and method the word “were” was repeated twice! You need to check the grammar and spelling multiple times before submission. 

No need to look further before the authors fix the manuscript. The authors need to fix it first before any considerations.

The figures are Okay.

The tables are good. 

References looked good. 

Comments on the Quality of English Language

Okay

Round 2

Reviewer 1 Report

Comments and Suggestions for Authors

Review 2nd round

Summary: The quality of the manuscript was improved. However, I still have some major remarks:

Major remarks:

I would like to revisit my first major remark from the first review:

1. The in vitro studies was carried out using a commercial kit. First question that comes to mind is why was the concentration range of T3 different between TRa and TRβ Especially since the potency for TRβ is higher, the lower amount results in a cut off before saturation is reached. Here, an estimation of potency is questionable. The difference in concentration range is briefly mentioned in the materials and methods section, but no further explanation was given. In order to fully interpret the data concerning the compounds, it would be helpful to include some kind of negative control/blank. Compounds without agonistic effect would be similar to this control.

Answer: It is true that the potency or the EC50 values may not be accurate enough in this study. However, there are significant differences in EC50 values among different methods (Ocasio and Scanlan 2008; Levy-Bimbot et al. 2012), thus, we do not consider accurate EC50 values critical for assessing the thyroid hormone activity of OH/diOH-PBDEs. The assay was validated in the presence of 1% DMSO. Figure 3 presents the results after subtracting the blank. Although we did not include a negative control in Figure 3, we can see from Figure 4 that, except for 6OH-BDE-47, the other OH/diOH-PBDEs did not exhibit thyroid hormone receptor activity activities. Therefore, OH/diOH-PBDEs except for 6OH-BDE-47 can be regarded as negative controls.

Reviewer answer: I agree that EC50 values might not be critical for the assessment the thyroid hormone activity (which was indirect in your study anyway), especially as . However, it is still unclear why two different concentration ranges were measured for TRa and TRb. Moreover, judging from figure 3A, it seems like there are data points for T3 values above 2000 nM, which was the maximal concentration mentioned in the text. Can the authors explain the choice of concentration ranges for the two different receptors and the data points above 2000 nM in figure 3A? I suggest to revisit the  material and method section. As for the negative control/blank I agree that displaying a any kind of negative control when blanks are subtracted does not add any information.

Figure legends: Although important information was added to the figure legend (mainly statistical nature), figures 3, 4 and 6 are still insufficiently labeled. Beginning with a title as it has been done for figure 5 for example. For figure 3, I suggest “Concentration-response curves of thyroid hormone receptor coactivator SRC binding assay”. Then explain what is seen in each segment (TRa in A and TRb in B). You could also mention EC50 values. In most studies, statistical information is shorten to “Values represent mean +/- SD (or SEM) from three independent experiments performed in triplicates” or similar phrasings. Afterwards the sentence about curve fitting or statistical analysis can follow.

Minor:

Line 52f: the addition of “ocean and bioaccumulation” produced an incomprehensible sentence

Line 57: please remove “were” to improve grammar

Line 89: This is the first time mentioning “thyroid hormone receptor” and abbreviation needs to be added here and can be used from then on. So the full term can be removed from line 91

Line 120: Please change “…utilizing the” to “…utilizing a” as you don’t specify which one you used

Comments on the Quality of English Language

Language quality has been improved immensely! I could only detect few grammatical errors as mentioned above.

Reviewer 2 Report

Comments and Suggestions for Authors

The authors seem to have gone through several revisions. Thank you for your effort, and I hope you'll make suggestions for future research. 

Comments on the Quality of English Language

Please, don't forget to double-check for typos.

Author Response

Future research should aim to deepen the understanding of the molecular mechanisms underlying the activity of OH/diOH-PBDEs and their interactions with thyroid hormone receptors, particularly focusing on their interactions with Helix 12. Understanding how these interactions influence ligand agonistic/antagonistic activity could provide valuable insights into the mechanisms underlying thyroid hormone modulation. Furthermore, cellular experiments and in vitro experiments associated with the thyroid system are necessary to elucidate the toxicity and exposure risk of OH/diOH-PBDEs.

Reviewer 3 Report

Comments and Suggestions for Authors

The edited/New version is much better than the old version. 

Author Response

We sincerely appreciate the constructive comments and insightful suggestions provided by the Reviewer.

Round 3

Reviewer 1 Report

Comments and Suggestions for Authors

Author Response

We sincerely appreciate the constructive comments and insightful suggestions provided by the Reviewer. Fig. 3 was added back to the manuscript. The limitation of T3 concentration range set and the accuracy of the related EC50 values were discussed in the manuscript: “Based on Fig. 3, the peak concentrations of T3 for TRα and TRβ approached but did not reach a maximum 520/495 signal. This outcome could potentially impact the precision of the EC50 values. However, despite setting the maximum concentration of OH/diOH-PBDEs in Fig. 4 close to their maximum solubility in water, a complete dose-response curve for OH/diOH-PBDEs was still not achieved. Consequently, conducting a comparison of EC50 values between OH-PBDEs and T3 is infeasible, and the relatively lower accuracy of the EC50 values for T3 can be considered acceptable.”

Round 4

Reviewer 1 Report

Comments and Suggestions for Authors

I thank the authors for taking my suggestions and believe the manuscript is now suitable for publication.